# Rabbi Nachman's Sonic Schemes

**Assaf Shelleg**

Department of Musicology, The Hebrew University of Jerusalem, Mount Scopus, Jerusalem 9190501, Israel; shelleg.assaf@mail.huji.ac.il

**Abstract:** This article discusses Tzvi Avni's Second Piano Sonata, *Epitaph*, a sonic commentary on one of the inner tales in Rabbi Nachman's "The Seven Beggars". Written between 1974 and 1979, *Epitaph* not only marks the composer's act of translation (from words into music and from a textual tale into a wordless and semantically unmarked piano sonata) but also his very turn to ethnographic sources that defied their negative function in a national territorial culture that vilified otherness while separating art from ethnography. Avni's turn to Rabbi Nachman was part of a bigger shift that saw composers' dialectical returns to Jewish histories and cultures that were previously repressed from a national culture which dehistoricized *the* Diaspora to the point of rendering the times and cultures of diasporic Jews a single temporality—ahistorical, contextless, and outside the teleological time of Zionism. With the (re)introduction of diasporic temporalities, non-redemptive poetics became an affordance in the music of Avni or Andre Hajdu (who is also discussed here) while steadily muting the territorial tropes that constituted Hebrew culture.

**Keywords:** Israel; Jewish music; art music; diasporism; Tzvi Avni; Andre Hajdu; Hebrew culture; Rabbi Nachman of Bratslav

The title is a metonym for the manner by which religion and theology have outgrown the receptacles allocated to them in the territorial project of Zionist Hebrew culture. More accurately, the title denotes a transition in the 1970s during which religion and theology ceased to serve a territorial function in Hebrew culture while becoming a modern—modernist, even—device in the hands of composers seeking dialectical returns to diasporic, non-sovereign histories and cultures outside the policed temporalities of Zionism. Rabbi Nachman of Bratslav in this regard embodies one such temporality that is at once non-territorial and non-redemptive. During the 1970s, options of this kind were signaled in compositions by Andre Hajdu whose setting of texts from the Mishnah and the Talmud had been animated by Jewish ethnographic imports that eschewed the modernist separation of art from ethnography, or by Tzvi Avni who commented on Rabbi Nachman's tale "The Seven Beggars" while reconfiguring melody types from the Jewish Eastern European soundscape (hence our title, which refers to the sonic schemes in Rabbi Nachman's tale but equally to Avni's commentary on them, in his Second Piano Sonata). Prior to getting into these works, however, a look at the networks wherein art, ethnography, and diasporism collapse into one another is in order, and all the more so since this fuzziness deems the secular veneer of Zionism not only porous but also friable. The portal through which this veneer was sustained saw actualized readings of the Bible, the colonization of its linguistic register, and the territorial function both had facilitated in Hebrew culture. Enter Zionist biblocentrism.

Zionist biblocentrism, the selective appropriation of biblical texts that reenacted Hebrew sovereignty through the Hebrew language (the single surviving linguistic component of ancient Jewish independence) conferred directionality to the Zionist project. Literal readings of selected biblical texts facilitated a Zionist national allegory that actualized tropes of return and returning, redemption, and territorial expansionism (Shapira 2004). The Bible validated the Zionist story by attesting to Israel's primordial ownership of the

land: It anticipated the return of the country's inhabitants ("And there is hope for your future—said the LORD—and the sons shall come back to their place [country]"; Jeremiah 31:16), the building of the land ("Yet will I rebuild you and you will be built, O Virgin Israel"; Jeremiah 31:3), and the gathering of exiles ("And I will take you from the nations and gather you from all the lands and bring you to your soil"; Ezekiel 36:24)[1]. At the same time, Zionism exalted the Bible by actively fulfilling its prophecies and confirming its eternal truth—be it the decree given to Abraham (Genesis 12:1–2), the exodus from Egypt, the feebleness of the generation sentenced to perish in the desert (Numbers 14:31–32), the stoutness of the conquerors of Canaan, the wars of the judges, or the institution of Israelite monarchy (Simon 1999). The realization of biblical sovereignty in the Zion of the twentieth century therefore shaped an imposing Zionist allegory; so much so that in a 1936 memorandum submitted to the Palestine Royal (Peel) Commission, which convened following the first stage of the Arab Rebellion, the Jewish Agency traced the origin of the association of the Jewish people with the land of Palestine in the "early pages of the biblical record", specifically Genesis 15:13–14, 46:4, 40:15; Leviticus 26:44–45; and Deuteronomy 30:3–5 (Jewish Agency for Palestine 1946). Transcribing the sacred into the secular allowed Zionists to conquer and monopolize the diachronic spaces of Jewish presence and absence in the land of Israel while stripping the biblical texts of exilic post-biblical rabbinic literature (Mishnah, Talmud, Midrash, Responsa, Zohar, Hasidic literature, etc.) that did not abide by territorial expansionism (Shaked 2004).

The national territory activated biblical and eschatological promises of return and (political) redemption while the musical settings of such texts rehearsed these national territorial topoi, irrespective of composers' stylistic and aesthetic penchants. It is therefore unsurprising to find both Erich Walter Sternberg and Paul Ben-Haim setting Ezekiel's Vision of the Dry Bones to music in the late 1950s; notwithstanding their distinctive compositional attitudes, both animated Ezekiel's vision, which the Zionist interpretative community had read and heard as an unequivocal metaphor for the restoration of Israel (much like the way post-biblical commentary, too, had interpreted it literally) (Hyman and Yitsḥak 2009). Verses like "The breath came into them and they lived, and they stood up on their feet, a very very great legion" (Ezekiel 37:10) or "Thus said the Master, the LORD: I am about to open your graves, and I will bring you up, My people, from your graves and bring you to Israel's soil" (Ezekiel 37:12) lent themselves quite naturally to Zionist literalist readings. Correspondingly, musicalized prophecies envisioning gaping graves and the slain breathed with life, growing flesh, and forming skins were belated extensions of the predominant paradigm of the living dead in modern Hebrew poetry, which a decade earlier had equated burial with espousal, and internment with sprouting and blossoming (Miron 2010; Hever 2014). Politically obedient and with limited command of Hebrew, a lack that among other things determined Ben-Haim's choice of this biblical text, his *Vision of a Prophet* (1958–1959)—a cantata for tenor solo, choir, and orchestra, which set Ezekiel Vision of the Dry Bones (Ezekiel 37: 1–14)—would amount to a triumphal monument cemented by (intervals of) fifths growing tonal skins and making audible the oncoming resuscitated marching multitude. Manifestations of this kind emerge most conspicuously in the setting of Ezekiel 37:11 ("Man, these bones are all the house of Israel") as the choir pierces this processional march with a secco, non-pitched *parlando* ("Our bones are dry and our hope is lost; we have been cut off"). Ben-Haim modulates this setting into choric shouts against the background of chromaticism he consolidates through glaring fanfares into an aria (for the tenor) to deliver the symbolic resurrection of the dead ("I am going to open your graves and lift you out of the graves, O my people, and bring you to the land of Israel"). Tonality at this moment is paired with redemption while affirming return *to* (and *of*) the land. Many such formulations would follow, all the way to the 1980s.

But all this did not prevent the gradual separation of Hebrew culture from the heteronomy of the territory (even if not yet not from the space in which these hybrids came to be). One such precursory instance is Abel Ehrlich's 1965 *Do not be like Your Fathers*, a setting of Zechariah 1: 4–5 for mixed choir, which seemingly checks all the "right" boxes. A biblical

text sung by a choir—the metonymic voice of the nation—polyphonically proclaiming the following dialogue (with verse 5 being the people's rejoinder to Zechariah): "Do not be like your fathers to whom the former prophets called saying, Thus said the LORD of Armies: Turn back, pray, from your evil ways and from your evil acts. But they did not listen to Me, said the LORD. Your fathers, where are they? And the prophets, did they live forever?" Except that the people's response is somewhat unsettling, or at least uncharacteristic compared with the settings of the Bible in the 1940s and 50s by composers like Ben-Haim, as his previously discussed *Vision of a Prophet* shows, or Ben Zion Orgad, who in 1949 positioned himself as the custodian of the nation's welfare and the consoler to the entire Jewish collective (as many poets, members of his cohort, had done) while setting David's lament over Saul and Jonathan (2 Samuel 1:19–27) as a political–theological drama for baritone and orchestra titled *The Splendor, O Israel* (*Hatzevi Israel*). (Other examples include Orgad's 1953 cantata Isaiah Vision or Haim Alexander's 1956 And it Shall Happen in Future Days [Vehaya be'acharit hayamim], both of which set Isaiah 2: 1–5, a text to which Ben Gurion, too, had given an emphatically eschatological meaning while underscoring its universalist stance) (Shelleg 2020; Ben-Gurion 1972). While works like Ben-Haim's *Vision of a Prophet* paired the militant with the redemptive and thus sought to actualize a biblocentric, literal reading of Ezekiel's Vision of the Dry Bones, they equally attested to willing participants who could not resist the high voltage of Zionist ideology. Ehrlich's textual choice, however, was equivocal enough to destabilize the biblocentric mechanism that sought actualization. When read literally, the verses from Zechariah 1:4–5 do not only oppose the way the biblical forefathers reinforced the territorial foundation of Hebrew culture; in the limited context of two verses, which is the equivalent of dozens of decontextualized appropriations that lent themselves to literal national readings of this kind ("Walk humbly" read the sign at the Haifa *Re'ali* School, thus excluding the subject of this sentence, which in the full text from Micha 6:8 refers to walking humbly *with God*), the people's reply to Zechariah suggests that they have become dismissive of the past. Yet, it is the same past from which Hebrew culture had culled its themes, diction, rhetoric, and myths. Ehrlich, moreover, further exacerbates this witty textual choice by using agogic accents on the opening verset ("do NOT be like your fathers"), thereby turning Zionist biblical literalism on its head.

Indeed, if the biblical text is stripped of its exilic, post-biblical strata, it could simply mean "do not be like your exilic fathers" (under the Zionist purview of rhetorically repressing Jewish diasporas), or even "do not be like the biblical fathers who were commandeered to constrain you to a national *telos*". And this becomes even more confusing come verse 5, which Ehrlich does not seem to read as the people's rejoinder. In his setting, the words "your fathers" are broken into extended syllables in a slowly unfolding polyphonic texture whose adjacent pitches produce a dense chromatic cluster suggestive of an ethereal or even ghastly imagery, implying that these "fathers" are simply (or literally) dead.

By the early 1970s, as ultra-right-wing religious nationalists maximalized the literalist manner by which the Bible was read in earlier decades and shifted it toward messianic territorialism in the newly occupied territories in the West Bank and the Gaza Strip (Shafir 2017; Perliger and Pedahzur 2021), composers sought other territories—territories that featured the ethnographic spaces in, around, and outside the State of Israel as well as the histories that preceded national identitarian earmarks. The abundance of incongruencies stemming at this point from Zionist biblocentrism, from the agency of oral Jewish musical traditions, and from the subsequent emergence of aesthetic hybrids stained by the theological infrastructure of Zionism, saw dialectical returns to Jewish diasporic and ethnographic imports that could no longer abide by territorial nationalism. Returns to diasporic Jewish cultures attenuated immediate national symbolism as much as they anticipated the faltering of national symbolization by their sheer pluralities. With the usurpation of diasporic Jewish cultures, national referents were progressively decentered, and Hebrewist signifiers ceased to activate the national soundboard. All this was not commensurate with the thrust of latent messianism in the rank and file of religious Zionism and the subsequent creeping

annexation of the occupied territories. Several prophetic texts from the early 1950s had warned about the radical consequences of too close an overlap between theology and politics (Simon 1982; Kurzweil 1971; Leibowitz 1992), but it took the demise of the (Ashkenazi) political elite in whose image Hebrew culture had been conceived, and the ideological vacuum this founding cohort found itself in during the 1970s, before religious Zionists could repurpose nationalized theological vessels and infuse budding messianic ambitions.

But composers looked elsewhere. Compositions written during the 1970s evinced turns to the narrative leftovers of the materials from which Zionist myths were culled. These featured post-biblical texts that were excluded from the national allegory, including returns to premodern Ashkenazi diasporic culture, which positioned itself as supra-ethnic. The person to spell out its ethnicization most violently was Andre Hajdu, a Holocaust survivor who fled Hungary in 1956 to live and work in Paris (where he studied with Darius Milhaud and Olivier Messiaen) prior to immigrating to Israel in 1966 and becoming a *baal teshuvah*. Hajdu's early European works betray a non-sublimated deconstructive approach that jeers at socialist realism (against whose dictates he received his early musical training) and equally at post-World War II avant-garde compositional approaches that were the lot of the West Bloc. Hajdu's modest compositional harvest from these years resounded the disbelief of an ethnomusicologist, who, having been trained with Zoltán Kodály in Hungary, could meticulously transcribe his wildest improvisatory impulses, and perhaps more importantly critically assess the road traveled by ethnographic imports as they become "artsy" constructs. In Israel, Hajdu's 1970 *Ludus Paschalis*—where Talmudic excerpts (see Figure 1) and psalmodic verses were gradually eclipsed and eventually stained by the singing of Christian children performing a mock Easter play—began to stage what Hebrewism had repressed. Having manifested the cultural and religious adjacencies of Jews and Christians while setting texts that undermined the Zionist management of Jewish history (including its vilification of Jewish diasporas), Hajdu's returns to Europe (as a composer and an ethnographer) and to Judaism (as an observant Jew) were preoccupied neither with (auto)exoticism nor with defamiliarization of post-World War II avant-garde. Following his ethnographic fieldworks in Hasidic Jewish communities and institutions for advanced Talmudic studies (*kolelim*) in Israel during the late 1960s, Hajdu could ethnicize, or rather *re-ethnicize*, Ashkenazi Jewry by returning to the early stages of Western musical literacy wherefrom he fleshed out repressed ethnographic and liturgical adjacencies.

*Ludus Paschalis* saw Hajdu's penchant for non-sublimated formulations that staged raw and unvarnished psychic energies in a *seemingly* unskillful heterophony of Jewish and Christian musics in late medieval times. With no aesthetic debts to Hebrew culture, its rhetoric, and territorial stipulations, Hajdu saw no breaks between the land of Israel and everything Jews had created in the Diaspora (Zakai and Hajdu 2015). The historicist mechanism at the basis of his writing, in other words, was not dissimilar to the Hebrewist return to origins (as was the Canaanites' idealization of a pre-monotheistic past, for example) (Ohana 2012), but the difference was crucial. Hajdu perceived both Judaism and Western existentialism as part of his "pioneer" vision; having realized that the Israeli elite had been decentered and disoriented, while exilic experiences were still too close and the "instinct to run away from them was strong",[2] he could now unload the imports that had been simmering within him since the late 1950s.

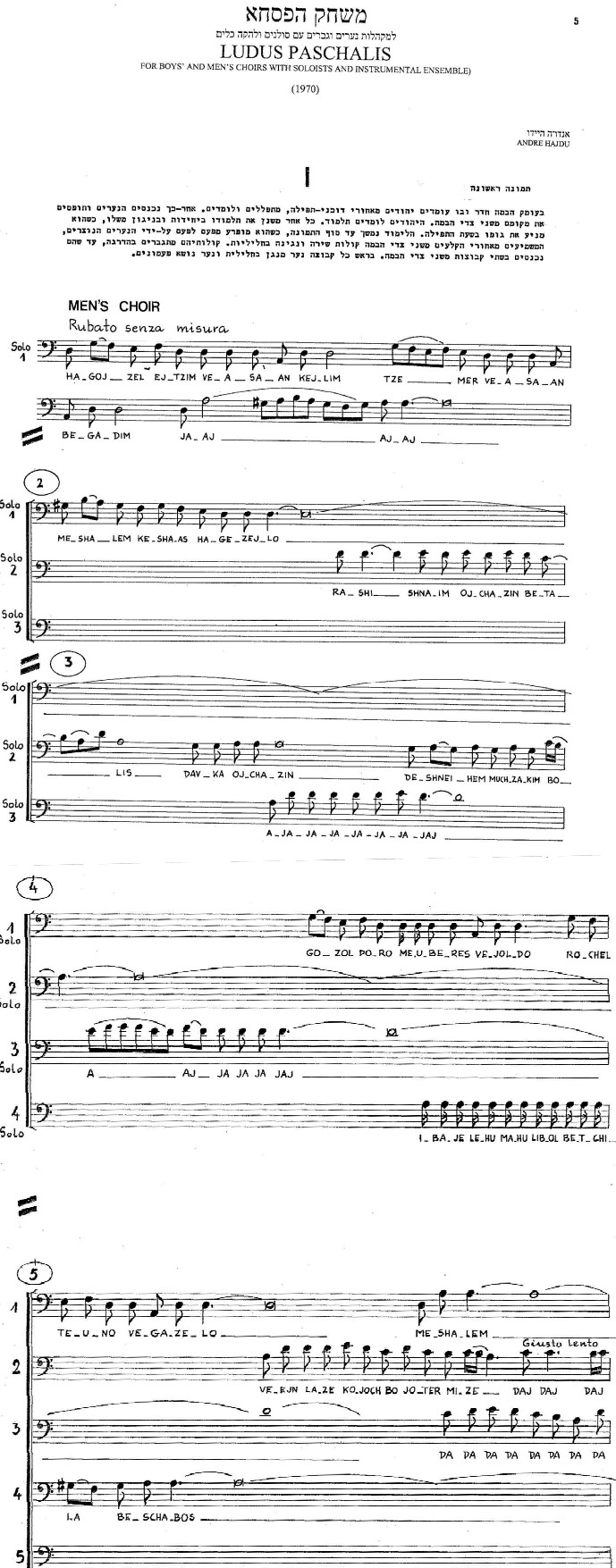

**Figure 1.** Andre Hajdu, *Ludus Paschalis* (1970), scene I, opening.

Set for a (Jewish) men's choir (with occasional flickering solo parts), and a (Christian) boys' choir whose members also play percussion instruments (bells, cymbals, whistle, rattle, bullroarer, small casserole, and side drums), Hajdu's *Ludus Paschalis* sought the ethnographic reproduction of the sonic spaces shared by Jews and Christians. A production inseparable from the autobiography of the composer–ethnomusicologist, *Ludus Paschalis* set tractates from the Babylonian Talmud alongside passages from medieval Passions and Easter plays that called for the participation of impersonated Jews while sampling the Holy Week violence. Hajdu's main source for such texts was Ernst August Schuler's 1951 *Die Musik der Osterfeiern, Osterspiele und Passionen des Mittelalters* (Schuler 1951) in whose spirit of mock or corrupt Hebrew he took the liberty to further decontextualize several words and refashion others while animating the Talmudic and Mishnaic text he had set with a *gemore-nign* (Talmud melody; see Figure 1), which he studied during his fieldworks in Israel in the late 1960s (Hajdu 1968; Hajdu 1971a; Hajdu and Mazor 1971; Mazor et al. 1974).[3] Hajdu's ethnographic attention was given to the unsublimated and ultimately violent energies stemming from such religious and sonic adjacencies. Having interpreted Zionist diaspora-negating rhetoric as disingenuous, given the gap he discerned between national rhetoric and the hybrids he had been exposed to during his fieldworks, Hajdu staged the repressed texts, musics, and temporalities of the Mishnah and the Talmud. Both were granted the status of motifs alongside the staging of unwanted Christian contiguities. In addition to the blatantly non-biblocentric narrative and the employment of an Eastern European (read: exilic) soundscape in the form of *gemore-nigns* (in numerous variants), *Ludus Paschalis* allowed neither Zionist diaspora-negating rhetoric nor "symptoms of a linear march toward intolerance" (in David Nirenberg's words) to be easily converted into territorial paradigms (Nirenberg 1996; Chinski 2015). Hajdu was the only person among his peers capable of transcribing such musics *about* Jews while being indifferent to—and disinterested in—national compulsions. *Ludus Paschalis*, as a result, bluntly disregarded the modernist separation of art from ethnography as much as it scorned the vilification of exilic Otherness and rendered the ethnicity of the Ashkenazi hegemony *relational* to those ethnic groups this hegemony placed on lower socioethnic rungs (which is also why *Ludus Paschalis* caused such a public kerfuffle immediately after its premiere) (Rosenblum 1971; Zilberman 1971; Bahat 1971; Bar 1971; Ezrahi 1971; Epstein 1971; Flusser 1971; Hajdu 1971b; Freier 1971; Ha'elyon 1971 (all in Hebrew)).

And with this we come to Rabbi Nachman. The following excerpt is taken from one of the parenthesized tales in his "The Seven Beggars," which is also known, separately (after having been popularized through its paraphrasing in the third act of Ansky's *Dybbuk*) (Ansky 1971), as the story of "the heart and the spring". This was the basis for the musical commentaries Avni had written between 1974 and 1979 and which came to be his Second Piano Sonata, *Epitaph*. Trained under Ben-Haim, Ehrlich, and Mordecai Seter at the Israel Academy of Music in Tel Aviv (1955–1958), Avni internalized Ben-Haim's (auto)exoticism but breached it all the same using the linear compositional devices of he saw in Ehrlich and Seter's works (as Avni's 1961 First Piano Sonata shows). But following his graduate studies at the Columbia–Princeton Electronic Music Center (1962–1964), Avni's employment of aleatoric compositional techniques saw the gradual emergence of a modal perception of twelve-tone writing with which he also sought the re-articulation of Eastern European melody types in the late 1960s. This collocation of contemporary techniques with Eastern European soundscapes (their disarticulation, more accurately) was signaled in *Five Pantomimes* for chamber ensemble and in *Jerusalem of the Heavens* for baritone, mixed choir, and symphony orchestra (both from 1968), but it appeared fully consolidated during the time *Epitaph* had been written—and all the more so given the sonic allusions embedded in Rabbi Nachman's text:

> Now there is a mountain. On the mountain stands a rock. From the rock flows a spring. And everything has a heart. The world taken as a whole has a heart. And the World's heart is of full stature, with a face, hands, and feet. Now the toenail of that heart is more heart-like than anyone else's heart. The Mountain

with the rock and spring are at one world, and the world's heart stands at the other end. The World's heart stands opposite the spring and yearns and always longs to reach the spring. The yearning and longing of the heart for the spring is extraordinary. It cries out to reach the spring. The spring also yearns and longs for the heart.

The Heart suffers from two types of languor: one because the sun pursues it and burns it (because it so longs to reach the spring); and the other because of its yearning and longing, for it always yearns and longs fervently for the spring. It always stands facing the spring and cries out: "Help!" and longs mightily for the spring...

Why doesn't the heart go toward the spring if it so longs for it? Because, as soon as it wants to approach the hill, it can no longer see the peak and cannot look at the spring. (When one stands opposite to the mountain, one sees the top of the slope of the mountain where the spring is situated, but as soon as one approaches the mountain, the top of the slope disappears—at least visually—and one cannot see the spring.) And if the heart will no longer look upon the spring, its soul will perish, for it draws all its vitality from the spring. And if the heart would expire, God forbid, the whole world would be annihilated, because the heart has within it the life of everything. And how could the world exist without its heart? And that is why the heart cannot go to the spring but remains facing it and yearns and cries out.

And the spring has no time; it does not exist in time. (The spring has no worldly time, no day or moment, for it is entirely above time.) The only time the spring has is that one day which the heart grants it as a gift. The moment the day is finished, the spring, too, will be without time and it will disappear. And without he spring, the heart, too, will perish, God forbid. Thus close to the end of the day, they start to take leave one from the other and begin singing riddles and poems and songs, one to the other, with much love and longing. This True Man of Kindness is in charge of this. As the day is about to come to its end, before it finishes and ceases, the True Man of Kindness comes and gives a gift of a day to the heart. And the heart gives the day to the spring. And again the spring has time. (Nachman of Bratslav 1978)[4]

The text is suffused with sounds—be it the heart crying for the spring, or both of them taking leave "singing riddles and poems and songs, one to the other, with much love and longing"—thereby making Avni's choice of this tale uncoincidental. Moreover, since the entire scene unfolds, subsists, and eventually extends through time, it suggests a chamber setting of a trio, as the heart, the spring, and the True Man of Kindness manifest a cosmic counterpoint that consists of its very balance. Rather than mimesis which would bill Rabbi Nachman's tale as the music's scenario, Avni's sonic commentary features abstractions; in lieu of programmatic portrayals, he opts for the semiotic reconfiguration of melody types that are part and parcel of the Eastern European Jewish soundscape—a reconfiguration that among other things manifests Avni's non-redemptive poetics.

And why non-redemptive? Because as long as Hebrew culture was conditioned by the national territory through actualized biblical promises of return (*to* and *of* the land), as long as these promises were complemented by the nationalization of theological tropes, and as long as the temporalities of the Hebrew language facilitated the centralization of the soil, Hebrew culture was bent on reproducing redemptive trajectories and a teleological directionality etched in territorialism.

Having read the transcribed version of the story (by Nachman's disciple and secretary, Rabbi Nathan of Nemirov whose remarks are recorded in parenthesis in the above-cited tale) and Buber's retelling of it (Buber 1956), Avni's choice of "the heart and the spring" was intertextually resonant; it leafed through the three versions of the story (Nemirov, Ansky, Buber) as well as through the sonic imagery animating the tale itself. Aware of

the dangers of signifying Nachman's symbolic fiction, Avni used the excerpted tale as the sonata's motto while circumventing mimesis; rather than "program music," he wrote in the notes to *Epitaph*, Rabbi Nachman's tale is "merely the spiritual starting-point—an idea in which more is concealed than is revealed, as it were" (Avni 1984).

Scholarly commentaries on "The Seven Beggars" and its sets of parenthesized stories are numerous. Joseph Dan maintains that the stories express the biography of the messiah and his endeavors in a folktale garb to bring forth redemption; at the same time, their web of texts, contexts, Kabbalistic myths, and lacunae discloses the secret autobiography of Nachman, which is "closely connected with an intensification of the centrality of his messianic destiny in his spiritual life" (Dan 2002a). Marianne Schleicher interprets "The Seven Beggars" as the collapse of a primordial past characterized by happiness, the pantheistic and panentheistic Hasidic principle; "When the infinite God decides to hand over creation to mankind," she writes, "the legitimacy of mankind as ruler in his world depends on man's recognition of his dependency on God". Because of man's inability to transcend time and space, however, he goes astray, failing to perceive his infinite divine; restrained by rationalistic perception, he "either chooses to take recourse to human wisdom or chooses simplicity" (Schleicher 2007). Since the children in the tale are too young to recourse to human wisdom and rationalism, they become beggars through God's intervention whose seven handicapped beggars embody man's limited perceptions; accordingly, the gifts bestowed upon the children by the beggars (who are crippled only in a corporeal world) lead to the annihilation of the gap between mankind and God.[5] Avni's Sonata focuses on one such instance, excerpted from the third beggar's (the stutterer's) homily. Fusing literature and mysticism and drawing on Kabbalistic and Hasidic concepts (Smith 2010), the segment about the heart and the spring occupies a central place in "The Seven Beggars". Dan finds that the very location of this tale "serves as the turning point in the narrative between the segments dedicated to the myth and of the creation and the primordial catastrophes and the segments leading toward correction and redemption. It is neither 'past' nor 'present,' but the description of an extended present, the situation in a universe that is in balance, even if a tenuous and temporary one" (Dan 2002b).

A corporeal world accustomed to worldly, profane speech renders the third beggar's praising of God and words of integrity the defective speech of a stutterer,[6] and yet it is he who prevents annihilation by sustaining the delicate and fragile interdependency of the world: he collects the true deeds of mercy by which time comes into existence, and hands them over to the True Man of Kindness (*tzaddik*). Meanwhile, the heart and the spring express their longing, love, and fear with the most beautiful riddles, poems, and songs, as if reenacting Psalm 61:3 ("From the end of the earth I call you. When my heart faints, You lead me to a rock high above me"), and in so doing offer the reader a variegated range of sonorities that evoke a "fugue of distance, love and timelessness", as Ora Wiskind-Elper's maintains (Wiskind-Elper 1998). The good deeds given by the True Man of Kindness (according to Schleicher) "recast the experience of prayer into a language that can bridge the unbridgeable gap" between the very being of this world (worshiper, heart) and the supernal world of infinity (God, spring), suggesting—as the intertext of Psalm 61 also indicates—that "prayers should be conceived of as music in the divine realm".[7] The tale therefore defines the theurgic relationship between righteousness and existence, bridged by the True Man of Kindness who gives the springtime, without which God would recede into pre-creation eternity and the world would cease to exist. As such, the third story in "The Seven Beggars" speaks of a fragile reality—a continuous static situation reaped unchanged and processual flows devoid of intrinsic crises that demand radical changes. Dan notes that the structure of the tale reflects Nachman's "concept of the present moment in cosmic history, in the movement between creation and redemption, according to the Lurianic messianic myth," yet it lacks the concept of redemption given that the processes unfolded in the story do not only describe a continuous, repeated unchanged, and static setting, but they also lack an intrinsic crisis that would demand radical change.[8] This non-messianic Kabbalistic concept is based more on the Hasidic doctrine of the role of the righteous (which

in the tale is one character divided into celestial and earthly entities) who embodies the intermediary power standing between humanity and God.[9]

Such combinations of mystery and fantasy, literature and mysticism, appearances and reappearances (read: variants and variations that are themselves echoes of non-Jewish folktales), timelessness, stasis (read: non-redemptive poetics), hidden meanings and mystical sonic imageries inform Avni's inventiveness. And like the tale itself, Avni moves from the revealed to the concealed: Timelessness is translated into non-metered proportional notation, while stasis is manifested through ostinati (continuous repetitions; see Figure 2), single notes (Figure 2, line 1), or aggregates (Figure 2, line 3). While such appearances and reappearances of melodic and harmonic ostinati were evident already in the second movement of Avni's 1968 *Five Pantomimes* (titled "Chagall, I and the Village", after the famous 1911 painting), in *Epitaph*, these devices abide by a hidden constellation, a concealed mode. As if following Nachman's allegorical mode wherein "the possibility of joining familiar, recognized elements in unexpected permutations," as Wiskind-Elper remarks,[10] Avni defamiliarizes and recontextualizes two Eastern European melody types—raised-fourth and *frigish* (Slobin 1980; Rubin 2020)—by displacing horizontal (melodic) or vertical (harmonic) projections thereof, to the point of rendering them disarticulated rather than signified. This was a conscious attempt to reconfigure ethnographic imports (however nucleus and not yet identifiable melodies) from which countless portrayals of Jewishness emerged—"'Samuel' Goldenberg und 'Schmuÿle'" from Mussorgsky's 1874 *Pictures at an Exhibition* (Taruskin 2009), Prokofiev's 1919 *Overture on Hebrew Themes*, Ernest Bloch's 1916 *Schelomo: Rhapsodie Hébraïque* (Móricz 2008), the 1964 *Fiddler on the Roof* (and even Cannonball Adderley's take on it from the same year), and numerous other instances. Avni reassembled the intervallic properties of the raised-fourth and *frigish* while most likely realizing that, from a scalar point of view, both are in fact rotations of one another. In the process, he embedded their most conspicuous and often stereotypical exotic marker—the augmented second interval—in a twelve-tone mode from which he could cull and project horizontal and vertical formations while continually using aggregates and registral displacements to dim the stereotypy of the augmented seconds and dilute the Eastern European melody types that house them.[11] Still, what stands behind such constellations if Avni's formulations are consistently inconsistent? And what sews the musical and the literary together here?

Avni's ostinati draw us closer to his reassemblage of Jewish musical markers as they maintain the tale's tension of attraction and existential separateness. Most ostinati occur on single notes—G♯, C♯, or B♭—revealing only partially the two augmented seconds onto which Avni grafts various projections of the raised-fourth and *frigish* melody types—shifted, transposed, foreshortened, and overlapped. Given the continuously partial horizontal or vertical projections of these melody types, however, they are constantly set in parallax, rendering each melody type a modal shift (or rotation) of the other. Subsequently, such changes in observational positions are less concerned with behavioral patterns of each melody type and more occupied with their relational and ensuing interdependency—an idea that is closer to Wiskind-Elper's characterization of "The Seven Beggars" as the "unremitting dynamic between the relative and the absolute: man's distance from God, his desire to draw near, and yet the impossibility of any union in life".[12] This ambivalence of attraction and separation in Avni's *Epitaph* records the return of the Eastern European soundscape (however veiled) in a modernist setting: Two melody types are embedded within a synthesized twelve-tone mode which continuously interrupts the stereotypy of the augmented seconds through melodic displacements and aggregates that de facto circumvent triadic harmony; in so doing, they defy the very separation of ethnography and art, and all the more so in a Zionist culture whose vilification of diasporic difference was necessary in order to eschew competing, non-sovereign territories as a viable option for Jewish national life (Boskovich 1953).[13] The three augmented seconds found in Avni's hybrid mode can therefore only allude to the raised-fourth or *frigish* melody types whose incomplete or displaced appearances become commentaries on their parallactic positioning. At the same time, each augmented second functions as a double leading tone that resolves

on one of four potential nodes (E, A, D, or G; see Figure 3) around which the entire sonata revolves. Looking back at *Epitaph* then, one realizes that almost all of the ostinati in the work occur on leading—and hence unstable—tones; as such, they become suggestive of the fragility of the ephemeral in Nachman's story which also accrues meanings from its surrounding parenthesized tales. The augmented seconds' parallactic distribution also explains Avni's choice to end the work on E, yet even this node (and pedal point) with which the work fades out is veiled by further displaced and fragmented allusions to the two Eastern European melody types. The sonata thus becomes a deliberate palimpsest of its own components.

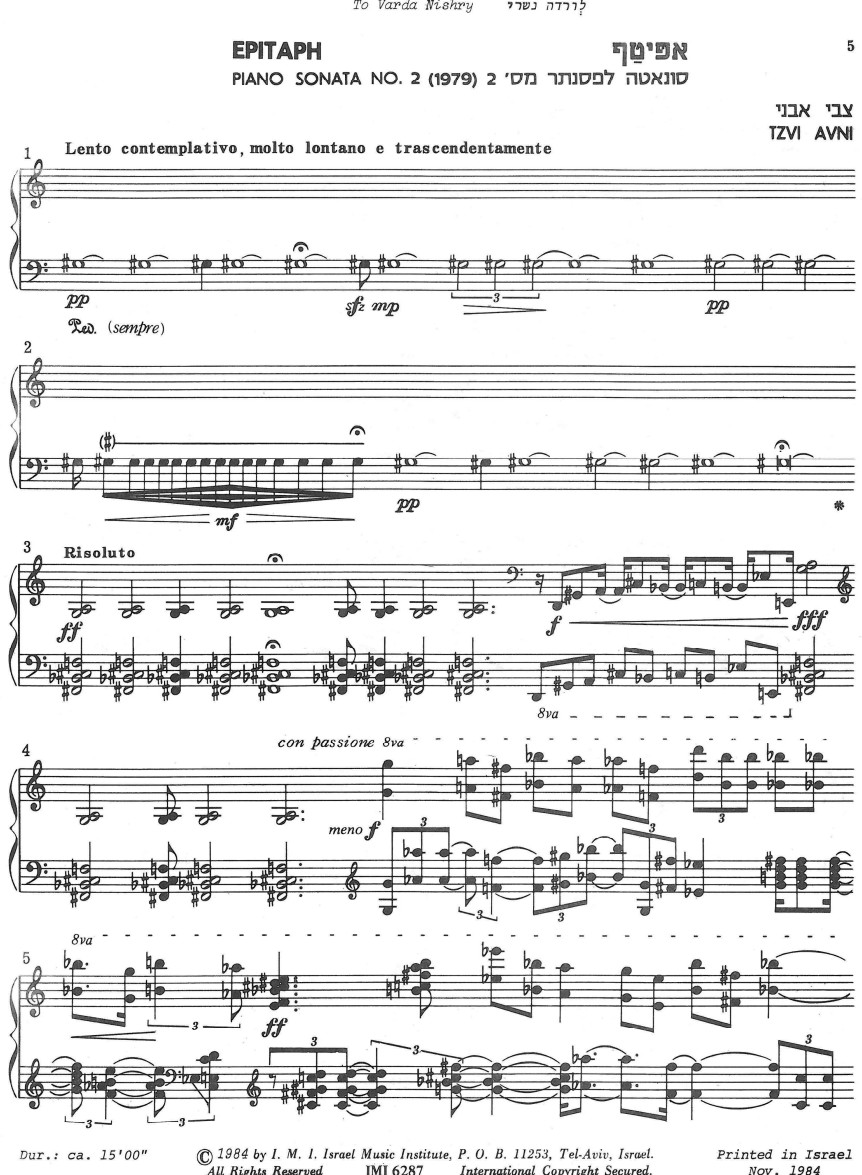

**Figure 2.** Tzvi Avni, Second Piano Sonata, *Epitaph* (1974–1979), opening (lines 1–6).

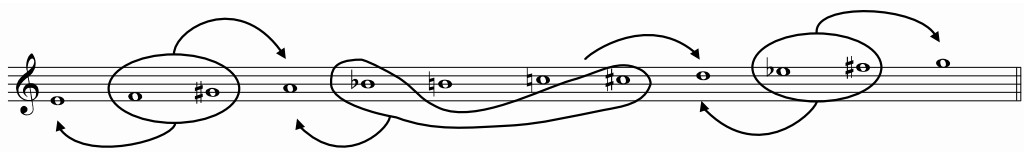

**Figure 3.** Avni's synthetic twelve-tone mode.

Nachman's symbolism freed Avni from motivic designs, but perhaps more importantly, it undercut the translation of Nachman's theurgy into redemptive trajectories. Avni's return to this diasporic text and temporality therefore lacked the teleology through which composers complemented the destruction–exile–redemption national (and hence territorial) paradigm, as in Seter's 1961 oratorio *Tikkun Hatzot* (*Midnight Vigil*), and equally so in the previously mentioned *Vision of a Prophet* by Ben-Haim (Shelleg 2023). To a certain extent, then, Hajdu and Avni's non-redemptive poetics (regardless of their stylistic choices and ethnographic imports) echoes Nachman's own turn to storytelling as a new mode of the instruction following the death of his infant son and the collapse of his messianic vision.[14] By the 1970s, therefore, the ideological validity of compositions that promoted territorial nationalism through exotic and/or neo-biblical imagery became a thing of the past, as musical formulations by Hajdu, Avni, and others not only ceased to function as ideological stand-ins but also signaled a transition from oppositionality to national tropes to a defiance of the modernist separation between ethnography and art (Latour 1993). In so doing, composers began to opt for diasporic nationalism that refused to reduce Jewish modernity to an ethnic–territorial dogma.

**Funding:** This research received no external funding.

**Institutional Review Board Statement:** Not applicable.

**Informed Consent Statement:** Not applicable.

**Data Availability Statement:** Not applicable.

**Conflicts of Interest:** The author declares no conflict of interest.

## Notes

1.     Translations of the Bible are taken from Alter (2019).

2.     Zakai and Hajdu, *Doors Opening*, pp. 80–81.

3.     See also Andre Hajdu Collection, MUS 173 A63, The National Library of Israel, Jerusalem. For Hajdu's ethnographic fieldwork recordings from the late 1960s, see National Sound Archives, Jerusalem, Y1391-1393, Available online: https://www.nli.org.il/he/items/NNL_MUSIC_AL990002437250205171/NLI (accessed on 3 March 2024).

4.     The parenthesized sentences in the cited passage are Nemirov's remarks.

5.     Schleicher, *Intertextuality in the Tales of Rabbi Nachman*, pp. 617–18.

6.     Schleicher, *Intertextuality in the Tales*, p. 586.

7.     Schleicher, *Intertextuality in the Tales*, pp. 588, 590.

8.     Dan, "Rabbi Nachman's Third Beggar", pp. 42, 47.

9.     Dan, "Rabbi Nachman's Third Beggar", pp. 47–48.

10.     Ora Wiskind-Elper, *Tradition and Fantasy in the Tales*, p. 215.

11.     At the opening of the piece, the ostinato on G♯ is led through an augmented second to F♮ while being submerged by the aggregate it flows into (Figure 2, lines 1–3). The aggregate itself displays some principles that govern the sonata's harmonic and melodic scheme: a verticalized raised-fourth tetrachord (G-A-B♭-C♯) framed by a major seventh (F♯-F♮), which would become a recurring intervallic reference in the sonata. The slightly shifted, quasi-heterophonic melody at the end of line 3 (Figure 2) also unfolds a raised-fourth tetrachord framed by an augmented seventh (E♭-E♮) that refers back to the first aggregate of line 3. Lines 4–5 (Figure 2) display a more complex two-part heterophonic texture from which several augmented seconds flicker: A♭-F in both the right and the left hand (their spelling as minor thirds notwithstanding) are followed by F♯-E♭ in the left hand. By line 5 (Figure 2), Avni displaces a *frigish* melody type (G-A♭-B♮-C-D) across the melody and its accompaniment that, once again, submerges an augmented second within a major seventh frame (Figure 2, line 5, second chord). A combination of a major seventh frame nesting an augmented second occurs in the middle of line 5 (which could be read as a permutation of the first aggregate from measure 3), yet it evolves into a post-tonal trichord (G-C♯-F♯) and an augmented second additive (C♯-B♭), thus harking back to a verticalized raised-fourth constellation. As the piece unfolds, the two melody types become intertwined while their auditory visibility is determined by the accompanying chords. Line 6, for instance (not shown in the example), demonstrates a shift from a raised-fourth allusion on F♯-G♯-A-C to another raised-fourth reference that extends from its last two notes into A-B-C-E♭. In line 11, a similar overlapping undergoes foreshortening: the melody unfurls D-E-F-G♯, transforms into G-F♯-E♭, and alludes to two *frigish* fragments (E♭-E♮-G and F♯-G-B♭) in the following bar, against a murky background of a "fist cluster" in the lowest register, thus erasing any sense of melody type and time.

12  Wiskind-Elper, *Tradition and Fantasy in the Tales*, p. 213.

13  Chinski, *Kingdom of the Meek*, pp. 280–87.

14  Smith, *Tuning the Soul*, p. 148. The writing of *Epitaph*, too, had been colored by the passing of Avni's mother and wife; the two women who consisted of his entire family died within two months of each other during the spring of 1973, thereby making the Eastern European soundscape (however modernized) a site of memory and loss.

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
