# Peer review of "Rabbi Nachman’s Sonic Schemes"

_religions, doi:10.3390/rel15040466_

Round 1

Reviewer 1 Report

Comments and Suggestions for Authors

You are commended -- celebrated -- for a highly original contribution to scholarship. I have three minor editorial suggestions:

1) line 19 - add Zionist Hebrew culture or, if you wish "Hebrew culture as inflected by Zionist territorial nationalism.

2) l. 154 -- receptacle is usually associated with rubbish; a receptacle for rubbish. It   would thus be best say "vessel."

3) Rabbi Nachman -- change to "Rabbi Nachman of Bratslav (1772-1810)

Comments on the Quality of English Language

The publication of this seminal article is a "feather in the cap" of Religions. Indeed, it is a unique contribution to scholarship, opening up new horizons for the study of religion.

Author Response

Many thanks for this review. I have implemented all suggestions. 

Reviewer 2 Report

Comments and Suggestions for Authors

The overall impression is that the article presents a very high level of mastery of the material and its immediate and wider context. The author's position is very clear.

There are few comments on the content:

In the lines 57-61 – “Transcribing the sacred into the secular allowed Zionists to conquer and monopolize the diachronic spaces of Jewish presence and absence in the land of Israel while stripping biblical texts of exilic post-biblical rabbinic literature (Mishnah, Talmud, Midrash, Responsa, Zohar, Hasidic literature, etc.) that did not abide by territorial expansionism” – the choice of terminology may be somewhat more general and less politicized (at least as it sounds), for example, “territorial issues (matters, considerations)” rather than “expansionism” in this context.

The balance between the musicological and cultural-political aspects of a theological (in essence) article is well maintained. That said, some of the terms used (e.g., heteronomy, metonym, teleological), while obviously familiar to cultural anthropologists, are probably not common enough among some music or religious studies scholars, who must still come to grips with phenomenology. Yet, this is not a request to change anything, except perhaps (1) to note "in a philosophical rather than a theological sense" for teleological in "the teleological time of Zionism" in the Abstract and (2) possibly to find a synonym for “heteronomy” (in line 92), where its meaning is somewhat blurred.

The technical comments are as follows:

·         The in-text reference numbers for endnotes should be superscript.

·         The musical example from Hajdu’s Ludus Paschalis is scanned crookedly (both in the supplementary file and as embedded in the text for peer-review). It would be recommended to rescan it.

·         Example 3 (in the peer-review version) is erroneously a reappearance of Ex. 2. And this should be the very last example (diagram of one musical line) from the supplementary file.

·         A typo was found in lines 434-435: "standins" - this perhaps should be "standings".

In conclusion: Minor changes are recommended in light of the above.

Author Response

Many thanks for this close reading. Heteronomy is meant here in the sense of "subjection to" as well as "governed by" territorialism. 

I will fix the crooked example and will replace the correct one in ex. 3. 

The typo will be corrected as well - it was supposed to be "stand-ins". 

Thanks again.